# The Role of Short-Chain Fatty Acids in Microbiota–Gut–Brain Cross-Talk with a Focus on Amyotrophic Lateral Sclerosis: A Systematic Review

**DOI:** 10.3390/ijms242015094

**Published:** 2023-10-11

**Authors:** Anca Moțățăianu, Georgiana Șerban, Sebastian Andone

**Affiliations:** 11st Neurology Clinic, Mures County Clinical Emergency Hospital, 540136 Târgu Mures, Romania; 2Department of Neurology, University of Medicine, Pharmacy, Science and Technology of Târgu Mures ‘George Emil Palade’, 540142 Târgu Mures, Romania; 3Doctoral School, University of Medicine, Pharmacy, Science and Technology of Târgu Mures ‘George Emil Palade’, 540142 Târgu Mures, Romania

**Keywords:** short-chain fatty acids, SCFA, gut microbiota, ALS, amyotrophic lateral sclerosis, gut-brain axis

## Abstract

Amyotrophic lateral sclerosis is a devastating neurodegenerative disease characterized by the gradual loss of motor neurons in the brain and spinal cord, leading to progressive motor function decline. Unfortunately, there is no effective treatment, and its increasing prevalence is linked to an aging population, improved diagnostics, heightened awareness, and changing lifestyles. In the gastrointestinal system, the gut microbiota plays a vital role in producing metabolites, neurotransmitters, and immune molecules. Short-chain fatty acids, of interest for their potential health benefits, are influenced by a fiber- and plant-based diet, promoting a diverse and balanced gut microbiome. These fatty acids impact the body by binding to receptors on enteroendocrine cells, influencing hormones like glucagon-like peptide-1 and peptide YY, which regulate appetite and insulin sensitivity. Furthermore, these fatty acids impact the blood–brain barrier, neurotransmitter levels, and neurotrophic factors, and directly stimulate vagal afferent nerves, affecting gut–brain communication. The vagus nerve is a crucial link between the gut and the brain, transmitting signals related to appetite, inflammation, and various processes. Dysregulation of this pathway can contribute to conditions like obesity and irritable bowel syndrome. Emerging evidence suggests the complex interplay among these fatty acids, the gut microbiota, and environmental factors influences neurodegenerative processes via interconnected pathways, including immune function, anti-inflammation, gut barrier, and energy metabolism. Embracing a balanced, fiber-rich diet may foster a diverse gut microbiome, potentially impacting neurodegenerative disease risk. Comprehensive understanding requires further research into interventions targeting the gut microbiome and fatty acid production and their potential therapeutic role in neurodegeneration.

## 1. Introduction

Amyotrophic lateral sclerosis (ALS) is a severe neurodegenerative disease characterized by progressive degeneration of motor neurons from the brain and spinal cord. This degeneration leads to a gradual decline in motor functions, often with rapid progression and no effective treatment available. Over the past few decades, there has been an observed increase in the prevalence of certain neurodegenerative diseases, including ALS. This increase could be attributed to various factors, including an aging population, improved diagnostic techniques, increased awareness, and last but not least, changes in lifestyle (diet, physical activity) and exposure to toxic environmental factors (certain toxins, heavy metals, pesticides) [1,2,3].

Over the past years, researchers have shown increasing interest in understanding the role of the gut microbiota, often referred to as the “second brain”, in various aspects of human health. This includes its potential impacts on neurodegenerative diseases [4,5]. The bidirectional communication between the gut and the central nervous system, known as the gut–brain axis (GBA), encompasses multiple signaling pathways. The gut microbiota stands out as a key modifiable factor that shapes the biochemical profile of the diet, thereby impacting neurodegeneration.

Gut microbiota can produce a wide array of metabolites, neurotransmitters, and immune molecules that can traverse into the brain, influencing the function of both neuronal and glial cells. Short-chain fatty acids (SCFAs) have garnered attention among the molecules produced by the gut microbiota due to their potential health benefits. Maintaining a diverse and balanced gut microbiota is crucial for producing SCFAs. A fiber- and plant-based diet can promote the growth of beneficial gut bacteria responsible for SCFA production [6].

Emerging research suggests that SCFAs play a role in influencing the GBA, potentially having effects on brain health and function. Modulation of the microbiota–gut–brain axis could prove to be a pivotal treatment strategy for neurodegenerative diseases in the context of the modern medicine era.

## 2. Methods

The primary objective of this systematic review is to explore the connection between ALS, the gut microbiota, and SCFAs. Other objectives included understanding how these factors can offer potential strategies for better understanding the different pathways involved in ALS, and how they can influence potential therapies in the future.

We undertook a systematic review following PRISMA (Preferred Reporting Items for Systematic Reviews and Meta-analyses) guidelines [7].

The search strategy included searches from two journal databases, NCBI PubMed and Google Scholar.

We performed a query using combinations of the following search terms: “SCFA”, “short-chain fatty acids”, “gut microbiota”, “brain”, “ALS”, “amyotrophic lateral sclerosis”.

No data or language restrictions were applied to the search engines. 

Studies on humans, experimental studies on animals, and in vitro studies were included. Duplicate findings were removed by manual revision.

After screening and duplicate removals, 134 studies were identified and included in our review (Figure 1).

## 3. Results and Discussion

### 3.1. Short-Chain Fatty Acids’ Synthesis, Metabolism and Relationship with the Gut–Brain Axis

Short-chain fatty acids (SCFAs) are aliphatic monocarboxylic acids containing one to six carbon atoms. They are produced through anaerobic fermentation of indigestible polysaccharides, namely dietary fibers, by the bacterial populations existing in the large intestine [8,9,10,11]. The most commonly encountered representatives are acetate (C2), propionate (C3), and butyrate (C4) [12]. Dietary habits and the gut microbiota primarily influence the quantity of each SCFA [9]. Uncommon sources of SCFAs include amino acid metabolism, which has the drawback of producing potentially toxic metabolites such as ammonia. Another source is butyrate synthesis, obtained from glycolysis-derived acetyl-Coenzime A [10,13].

After their production, SCFAs are predominantly absorbed by colonocytes through active transport mediated by monocarboxylate transporters (MCTs) [14]. Most SCFAs enter the citric acid cycle within the mitochondria, providing energy for cellular use. Another portion enters the portal circulation and serves as an energy substrate for hepatocytes, except for acetate, which is utilized for cholesterol and fatty acid synthesis [9,15,16]. Consequently, only a small quantity of SCFAs are available in systemic circulation [9].

Studies have demonstrated that brain–blood barrier (BBB) endothelial cells contain MCTs that facilitate BBB permeability to SCFAs [14]. Despite the ability of SCFAs to traverse the BBB, their brain uptake appears negligible [17]. Nevertheless, SCFAs play a significant role in the pathogenesis of various neurodegenerative disorders by locally and systemically modulating the gut–brain axis (GBA).

### 3.2. Short-Chain Fatty Acids’ Local (Intestinal) Effects

SCAFs are essential in maintaining and even improving the proper functioning of the gut. They achieve this by enhancing the integrity of the intestinal barrier, stimulating mucus production and gastrointestinal tract mobility, as well as exerting anti-inflammatory and anti-tumorigenic effects [18,19,20]. For instance, butyrate has demonstrated its value in strengthening the tight junctions between the intestinal cells, thereby preventing the harmful paracellular leakage of pathogens and potentially dangerous contents within the lumen [21]. 

Furthermore, butyrate not only stimulates the expression of the MUC2 gene, which is responsible for mucus production in the gut, but it also has the potential to modify the structure and the quality of mucin. As a result, SCAFs play a paramount role in protecting the intestinal mucosa against luminal substances such as pathogens, chemicals, and toxins [22]. 

Beyond its barrier function, mucus serves as a nutritive resource for specific microbiological species [23]. SCFAs have also demonstrated their role in shaping the inflammatory and immune environment of the gastrointestinal tract. Notably, butyrate possesses anti-inflammatory features by inhibiting mitogen-activated protein kinase (MAPK) pathways and nuclear factor-kB, resulting in a reduction of pro-inflammatory mediators such as tumor necrosis factor α (TNF α), interferon γ (IFN γ), interleukin (IL)-1, IL-2, IL-6, IL-8 [24]. 

Furthermore, butyrate stimulates the production of glutathione, thereby reducing the anti-inflammatory response triggered by reactive oxygen species. Butyrate also affects innate and adaptive immune systems by exerting anti-inflammatory effects. It does so by impeding the recruitment and pro-inflammatory activity of macrophages, dendritic cells, and effector T cells while simultaneously stimulating the regulatory T cells (Treg) population [25]. Lastly, SCFAs possess anti-tumourigenic features. They stimulate differentiation and cell death and create an inhospitable environment for the development, proliferation, and migration of tumor cells [19].

The gut–brain axis defines the interconnected relationship between the gut microbiota and the central nervous system. On one hand, the signals from the brain influence the sensory, motor, and secretory functions of the gastrointestinal tract. On the other hand, the microbiological population typically residing in the gut and their main metabolites, SFCAs, can modulate brain function through different pathways, including immune, endocrine, and neural routes [26].

Two major mechanisms underpin the SCFA-mediated interaction between the gut and the brain. Firstly, SCFAs function as endogenous ligands for various G protein-coupled receptors (GPCRs), with GPR43 and GPR41 (also known as FFAR2 and FFAR3) being the most extensively studied. Additionally, butyrate-sensing olfactory receptors are dispersed among the cells involved in the gut–brain axis, including neurons, immune cells, and enteroendocrine cells. Activation of these receptors leads to actions specific to their localization [27,28].

Furthermore, another crucial mechanism involves inhibiting histone deacetylase (HDACs) activity. This inhibition results in a larger quantity of transcriptionally active chromatin and significantly enhanced gene expression in targeted cells [29].

### 3.3. Immune Pathways

An organism’s immune and inflammatory statuses are closely intertwined with the pathogenesis of various neurodegenerative disorders. These connections involve interactions among pro-inflammatory molecules, immune cells, and neural pathways [30]. 

As mentioned earlier, SCFAs derived from the gut microbiota establish a local anti-inflammatory environment by reinforcing the intestinal barrier. This reinforcement limits bacterial translocation, modulates the maturation of diverse immune cells, and reduces pro-inflammatory markers by inhibiting different molecular pathways [31]. 

Given the demarcation between gut and systemic immunity relies on a single layer of epithelial cells, SCFAs can easily exert control over the peripheral immune system, leading to direct repercussions for the central nervous system (CNS) [32]. Strengthening the intestinal barrier’s integrity reduces systemic inflammation [9]. 

Moreover, SCFAs play a pivotal role in modulating both innate and adaptive immune systems. Notably, neutrophils, which are the first to arrive at the site of injury and mobilize other immune cells through cytokine secretion, are vital components of the innate immune response. SCFAs exert their influence via HDAC inhibition, thereby regulating the production of inflammatory cytokines and chemokines. In addition to this molecular process, SCFAs bind to FFAR2 receptors located on neutrophils, altering their chemotactic properties [33].

Regarding the adaptive immune response, SCFAs play a role in preventing the transformation of monocytes into macrophages and dendritic cells within the damaged tissue via HDAC inhibition. They also disrupt the ability of phagocytic cells to capture antigens and generate pro-inflammatory molecules [34]. SCFAs also influence T-cell development. Mainly, butyrate interacts with GPR109A located on dendritic cells, indirectly stimulating the production of different enzymes (e.g., indoleamine 2,3-dioxygenase 1, aldehyde dehydrogenase 1A2) with immunosuppressive features. This process leads to the maturation of naïve T cells into T regs [35]. Moreover, the direct inhibition of HDAC stimulates mTOR activity, producing T helper (Th)1 and Th17 cells [36].

Oral administration of SCFA reduces neutrophil recruitment and inflammation in various experimental models of induced colitis and T-cell dependent colitis. Notably, neutrophil recruitment and inflammation were decreased in mice with induced colitis who received butyrate enemas. It remains uncertain whether this reduction directly stems from SCFAs’ impact on neutrophils or if it is an indirect effect. SCFAs influence the production of reactive oxygen species by neutrophils, and their phagocytic capability against microorganisms and particles. Both acetate and butyrate result in an increase in reactive oxygen species production. Additionally, SCFAs modulate the production of inflammatory mediators, such as cytokines and chemokines, by immune and non-immune cells. SCFAs are shown to decrease the production of TNF-α [34].

Interactions between the microbiota and the immune system are bidirectional. Intestinal epithelial cells capture SCFA through both passive and active mechanisms. Once within the cells, SCFAs serve as an energy source and also enhance the expression of antimicrobial peptides secreted on the surface of epithelial cells. They additionally modulate the production of immune mediators, including IL-18. SCFA regulates the differentiation, recruitment, and activation of neutrophils, dendritic cells, macrophages, and T lymphocytes. Generally, they exhibit anti-inflammatory effects, such as reducing the production of TNF-α and IL-12 by macrophages and dendritic cells. Furthermore, SCFAs alter cells’ capacity to capture antigens and stimulate T cells [37].

Beginning with the hypothesis that fatty acids modulate macrophage function—the most prevalent immune cell in the lamina propria of the intestinal mucosa—an experimental model was employed. In this model, macrophages derived from bone marrow were stimulated with lipopolysaccharides and exposed to butyrate, propionate, or acetate. The subsequent measurement focused on the secretion of nitric oxide and pro-inflammatory cytokines. The outcome revealed that among the SCFAs, butyrate exhibited the most potent action. Notably, nitric oxide, IL-6, and IL-12p40 levels were considerably reduced in a dose-dependent manner in the presence of butyrate. No significant effects were observed on other pro-inflammatory cytokines, such as TNF-α or MCP1.

Interestingly, these effects remained unaffected by the GPR receptors that SCFAs are known to influence. Instead, these mechanisms seem to be linked to the inhibitory effect of butyrate on HDACs. This could prove useful in reducing pro-inflammatory effectors in the lamina propria, potentially mitigating exaggerated responses seen in pathologies like ulcerative colitis or Crohn’s disease [34].

The connection between these compounds and systemic inflammation biomarkers in humans was examined in a review focusing on the effects of SCFAs determined using prebiotics or synbiotics regardless of the administration route. Meta-analysis findings indicate that prebiotics lead to a reduction in C-reactive protein (CRP), while synbiotics bring about reductions in both CRP and TNF-α. Moreover, a noteworthy observation emerged: rectal acetate administration via an enema significantly reduces systemic inflammation. Similar results are observed when acetate is administered at the distal colonic level [38].

Administration of butyrate to mice produced by commensal microorganisms during starch fermentation facilitated the generation of extra-thymic Treg cells. Additionally, propionate, but not acetate, potentiated the de novo generation of Treg cells. These findings imply that bacterial metabolites facilitate communication between the microbiota and the immune system, influencing the equilibrium between pro- and anti-inflammatory mechanisms [39].

Butyrate produced by intestinal microbiota regulates the functions of colonic Treg and T CD4^+^ cells. While the action of SCFAs through surface receptors like GPR 43 has been reported to contribute to some functions, this is unlikely to be the mechanism of butyrate’s effect on Treg cells. Notably, acetate, a potent GPR 43 ligand, did not induce Th CD4 differentiation into Treg cells. The suppression of the butyrate transporter in the colonic mucosa of patients with inflammatory bowel disease (IBD) has been observed. Furthermore, a reduction in butyrate-producing bacteria within the intestinal microbiota of IBD patients suggests that butyrate deficiency could be implicated in IBD pathogenesis. Supporting this notion, butyrate enemas have been shown to improve colonic inflammation in IBD patients, though their underlying pathophysiological mechanism remains incompletely understood [35].

Butyrate has potent effects on various functions of the colonic mucosa, including inflammation and carcinogenesis inhibition, reinforcement of specific components of the colonic defense barrier, and reductions in oxidative stress. The mechanism involves inhibiting the activation of NFKB and histone deacetylation. The effects observed vary based on the concentration and models used, although human data are limited [40].

Microglia-generated neuroinflammation is now recognized as a hallmark of neurodegenerative diseases. Microglial cells, prominent components of the CNS immune system, play a crucial role in shaping appropriate neural circuits [41]. Studies conducted in mice have revealed that SCFAs possess the capacity to stimulate the normal structural and functional development of microglial cells [42].

Alterations in the microbial ecosystem, such as those resulting from antibiotic use, lead to changes in microglial morphology that promote neuroinflammation [43]. Moreover, supplementation with SCFAs promotes microglial cells’ anti-inflammatory and neuroprotective behavior, most likely achieved through the acetate and butyrate-induced inhibition of HDAC [10,44].

Global deficiencies in microglia properties and an immature phenotype associated with poor immune response were observed in germ-free mice. The microbiota’s limited complexity also led to defective microglia. Partial restoration of microglia properties was observed after recolonization with complex microbiota. Notably, mice lacking the FFAR2 receptor for SCFA displayed similar microglial defects, suggesting that the microbiota plays a role in regulating microglia maturation. Additionally, complex microbiota can partially correct microglial dysfunction [42].

### 3.4. Hormonal Pathways

Two specialized neuroendocrine intestinal epithelial cell types secrete gut peptides on the basolateral side in close proximity to blood vessels and afferent fibers that innervate the intestinal mucosa. Firstly, the enteroendocrine cells (EECs) constitute the largest endocrine organ in the body and are distributed throughout the entire gastrointestinal tract, with their density increasing distally [45,46]. EECs are notably equipped with highly expressed chemosensory receptors machinery, comprising GPCRs and nutrient transporters. This enables them to detect and respond to the intestinal environment, leading to the release of various peptides, including cholecystokinin (CCK) and intestinal peptide YY (PYY), as well as the incretin hormones glucagon-like peptide-1 (GLP-1) and glucose-dependent insulinotropic polypeptide (GIP) [45,46,47]. These incretin hormones are released in response to a surge of nutrients during a meal, subsequently stimulating insulin secretion [48,49].

The second type of neuroendocrine intestinal cell, known as the enterochromaffin cell (ECs), is responsible for producing approximately 95% of the body’s serotonin [5-hydroxytryptamine (5-HT)]. This serotonin acts on 5-HT receptors expressed on local gut neurons [47,50]. Gastric endocrine X/A-like cells (also called P/D1 cells) are accountable for the production of ghrelin, which binds to its receptor known as the growth hormone secretagogue receptor 1a (GHSR1a), now also recognized as the ghrelin receptor (GRLN) [51,52].

Gut peptides induced by nutrients can exert their effects through two distinct mechanisms. Firstly, they can act in a paracrine manner by activating vagal afferent fibers that innervate the intestinal epithelium. These fibers then transmit signals to the brainstem in the nucleus tractus solitaries (NTS), which subsequently relays signals to higher-order brain regions, such as the arcuate nucleus. Alternatively, gut peptides can also enter circulation and directly affect the brain in an endocrine fashion, transmitting signals directly to the NTS in the brainstem [53,54].

SCFAs bind to specialized receptors GPCR43, GPCR41, and olfactory receptor 558, which are localized on EECs and regulate EEC activity [55,56]. SCFAs induce the release of gut hormones such as GLP-1 and PYY, as well as γ-aminobutyric acid (GABA) and serotonin (5-HT), through a pathway that is at least partially dependent on the GPR43 receptor [55,56,57,58]. Notably, the gut microbiota acting through SCAFs significantly influences serotonin production. Butyrate, for instance, is recognized to stimulate 5-HT secretion by upregulating colonic mRNA levels of tryptophan hydroxylase 1 in ECs [59,60]. Germ-free mice exhibit lower serotonin levels compared to conventional mice, and these reduced levels are reversed upon gut microbiota transplantation [61].

#### 3.4.1. Glucagon-Like Peptide-1 and Peptide YY

Administration of propionate and acetate in cell cultures extracted from mice resulted in increased GLP-1 secretion in control mice compared to mice with FFAR2 deletion. Notably, in mice lacking FFAR2, the response to propionate was diminished, and the response to acetate was completely abolished. A similar trend was observed in mice with FFAR3 deficiency, albeit to a lesser extent. Furthermore, mice with FFAR2 deficiency exhibited a lower concentration of GLP-1. The basal level of active GLP-1 was reduced in mice with FFAR2 deficiency and in those with FFAR3 deficiency, albeit to a smaller degree. These findings highlight the significant influence of the activity of these receptors on GLP-1 secretion [58].

Oral administration of sodium butyrate to mice significantly elevated plasma levels of GLP-1 and GIP and moderately increased peptide YY (PYY) levels. Sodium propionate administration also increased GIP, while GLP-1 or PYY remained unaffected. In contrast, sodium acetate administration did not lead to similar modifications in gut hormone secretion. Interestingly, GLP-1 and PYY levels were observed to increase in the portal venous system, and their precursors were also detected in the ileum and colon. Despite the strong correlation between leptin and GLP-1, it is noteworthy that an increase in GLP-1 secretion due to a rise in SCFAs’ value can potentially enhance leptin’s action [62].

Cani et al. investigated the impact of prebiotics on gut microbiota and their correlation with the secretion of GLP-1 and PYY. Despite the relatively small sample size (n = 10), their study demonstrated that prebiotic treatment notably elevated hydrogen excretion in the breath and reduced appetite. Notably, GLP-1 and PYY levels measured 60 min after a meal exhibited a significant increase in patients who received prebiotic treatment [63].

Chambers et al. conducted a study to examine the effects of propionate on energy metabolism and the concentration of GLP-1 and PYY. After administering 10 g of inulin ester daily over 24 weeks, the study observed a reduction in the intraabdominal distribution of adipose tissue and a decrease in intrahepatocellular lipid content. Furthermore, the study demonstrated that intracolonic administration increased GLP-1 and PYY levels [64].

PYY is produced in the colon, particularly in areas with high concentrations of SCFAs resulting from fiber fermentation by the gut microbiota. GLP-1 and PYY exhibit an immediate increase after consuming digestion-resistant fibers, a phenomenon demonstrated in experimental studies involving both mice and human subjects. This response is attributed to various mechanisms of action, including the activation of the FFAR2 receptor. Activation of this receptor triggers an acute elevation in gut hormone secretion and leads to increased enteroendocrine cells, particularly those responsible for producing GLP-1 and PYY [55].

FFAR2 receptor-deficient mice were compared to a control group in an experimental study. The study revealed that propionate administration resulted in a substantial increase in the plasmatic levels of GLP-1 and PYY, observed both in the jugular and portal veins for the control group. However, this effect was significantly diminished in subjects with FFAR2 receptor deficiency. These findings demonstrated that FFAR2 deficiency can alter gut hormone production induced by SCFAs [57].

Building on the hypothesis that the SCFAs produced through the colonic fermentation of dietary fibers play a protective role against diabetes, Tarini et al. conducted a study investigating the impact of inulin on postprandial glucose, insulin, and gut hormone levels. The study found that consuming 24 g of inulin led to a significant increase in GLP-1 levels after 30 min. This increase was also associated with a negative feedback response that influenced the plasmatic concentration of ghrelin. Furthermore, the study observed a substantial rise in colonic SCFAs production immediately after fiber ingestion, accompanied by a postprandial decrease in free fatty acids. Notably, the suppression of fatty acids was temporary, as a rebound effect appeared later, returning levels to their pre-ingestion state [65].

In a study involving human subjects, two groups of patients were examined: one comprising normal-weight individuals, and the other comprising overweight/obese individuals. The study aimed to explore the impact of acute SCFA administration on gut hormone levels. After an overnight fast, the patients were given different beverages; the control group received water with glucose, while the other groups received water with glucose, 24 g of inulin, or 28.2 g of resistant starch. The findings indicated that neither inulin nor resistant starch resulted in a significant reduction in ghrelin levels during the initial 4–6 h when compared to the control group. However, after 6 h, the group consuming inulin exhibited notably lower ghrelin levels than the control group. The study also noted a significant increase in fatty acid levels among patients who received inulin but not resistant starch compared to the control group.

Interestingly, no discernible differences were observed between the overweight/obese patients’ group and the normal-weight group. In conclusion, the acute elevation of colonic SCFAs did not appear to influence the response of gut hormones like GLP1 or PYY. However, it did lead to a reduction in ghrelin levels [66].

In a double-blind, randomized study, researchers investigated the effects of colonic administration of SCFA mixtures on energy metabolism. The study revealed that SCFA mixtures led to increased fat oxidation, elevated energy consumption, and higher PYY levels. Additionally, they reduced lipolysis in overweight patients. The concentrations of SCFA mixtures used in the study were comparable in both ratio and value to those achieved after fiber consumption. This mechanistic insight could aid in comprehending the potential long-term advantages of SCFA administration in terms of body weight control and insulin sensitivity, particularly among overweight patients with resistance [67].

#### 3.4.2. Leptin

Leptin is synthesized in white adipose tissue, a significant endocrine tissue that produces hormones called adipokines. These hormones primarily target the hypothalamus [68,69]. The GPR1 receptor (also known as FFAR3) has propionate acid as a specific ligand. Activation of the GPR41 receptor increases leptin release and expression [70,71,72]. Both propionate and acetate stimulate GPCR43 (also known as FFAR2) in differentiated adipocytes, leading to fat accumulation. Acetate demonstrated higher selectivity for GPCR43, whereas butyrate exhibited greater activity for GPCR41. However, propionate remains the most potent agonist for both receptors. The GPCR41 receptor (also known as FFAR3) has been noted for its high expression in adipose tissue, particularly in white adipose tissue, whereas its presence in brown adipose tissue is less detectable. Leptin, an intestinal hormone, is recognized for its potent ability to suppress appetite [73]. 

Acute propionate administration in murine models resulted in a twofold increase in plasma leptin concentration. However, this change did not significantly alter the difference in food intake between the propionate-treated and the control groups. Interestingly, studies conducted on other animal species, such as chickens or sheep, revealed a noticeable effect on food intake following increased leptin levels. This discrepancy suggests the presence of race-specific mechanisms at play. The concentration of SCFAs required to stimulate the elevation of leptin levels falls within physiological limits, indicating the potential relevance of this pathway for future studies in humans [72].

Propionic acid stimulates leptin expression and mitigates resistance in adipose tissue. This implies that propionate plays a role in energy and inflammatory metabolism, potentially influencing the development of obesity and type 2 diabetes. Notably, propionate significantly enhances messenger RNA expression of leptin while not affecting adiponectin. Additionally, propionate reduces messenger RNA expression of resistance. Interestingly, messenger RNA levels of GPCR 41 and GPCR 43 were significantly higher in subcutaneous adipose tissue compared to the omentum [74].

An unrecognized factor contributing to obesity is increased resistance to leptin. Modifying the gut microbiota composition using probiotics has shown an enhanced sensitivity to leptin in mice with diet-induced obesity and type 2 diabetes mellitus. Additionally, this alteration increased the available concentration of SCFAs. Notably, this change also led to a reduction in the inflammation status, which in turn improved leptin sensitivity [62].

#### 3.4.3. Other Effects

In another experimental study conducted on mice, the animals were divided into two groups: one with deletion of the GPCR1 receptor gene and a control group. The first group exhibited a significantly lower weight and a leaner constitution. Both groups were then colonized with two types of bacteria from the Saccharolytic bacterium family, typically found in the human intestine. Additionally, two identical groups, mirroring the first two, were not colonized with this bacterium. Interestingly, the differences observed between the groups of mice that were not colonized with GPR41 deletion and the non-deletion colonized group were not as significant. This finding indicated that the GPCR41 receptor plays a role in regulating energy balance, but its effects depend on the presence of intestinal microbiota [75]. In mice with FFAR3 deficiency subjected to a rich fatty diet, it was proven that administration of butyrate and propionate limited weight gain despite an increase in food consumption compared to the control group. This strongly suggests that SCFAs can effectively mitigate diet-induced obesity [76].

In a probiotics review, Le Blanc et al. highlighted various isolated bacterial strains capable of producing metabolites beyond SCFAs through food fermentation. Among these strains, *L. rhamnosus GG*, naturally present in the adult human gut microbiota, has shown the ability to synthesize vitamins B1, B2, and B9 in culture media. While the levels of these synthesized vitamins remain low, this strain stands out for its unique capacity to produce thiamine. Another bacterium, Bifidobacterium lactis BB12, found in dairy products, is one of the well-documented probiotics that exhibits biosynthesis potential for vitamin B1, though not for other B vitamins. The review also revealed several strains with reported B vitamin production abilities, albeit with significantly lower capacity. The review concluded that these specific bacterial strains might impact energy metabolism by enhancing ATP production. Additionally, they could play a role in investigating the relationship between chronic fatigue syndrome and alterations in the gut microbiota [77].

### 3.5. Neural Pathways

Beyond immune and endocrine pathways, the neurohumoral mechanisms through which SCFAs influence brain structure and function are equally significant. They can improve BBB integrity, regulate levels of diverse neurotransmitters and neurotrophic markers, modulate the brain neurochemistry, and, importantly, exert direct and indirect effects on the vagal nerve, which is an essential neuroanatomical link between the gut and brain [78].

The vagus nerve (VN) boasts the widest distribution throughout the human body, originating in the brainstem and extending down to the gastrointestinal system, with its nerve branches reaching all parts except the rectum. Through its sensory and motor branches, the VN plays a crucial role in vital functions like breathing, cardiovascular activity, and digestion [79]. It operates as a parasympathetic nerve, encompassing both afferent and efferent fibers. The afferent fibers constitute 80% of the VN and serve distinct functions based on their morphology and location within the intestinal layers [80].

The intraganglionar laminar endings in the myenteric plexus function as tension receptors. The intramuscular arrays in the muscle layer act as stretch or length receptors. The mucosal endings emerge around the intestinal crypt lumen or atop the villi, serving as chemoreceptors that can synapse with enteroendocrine cells using glutamate as a neurotransmitter [81].

Various gut hormones, including ghrelin, PYY, cholecystokinin (CCK), and GLP-1, interact with these afferent fibers, thereby influencing food intake and overall energy regulation [82]. The afferent fibers ultimately converge in the nucleus tractus solitaries (NTS) located in the medulla oblongata. The NTS is closely connected with the dorsal motor nucleus of the vagus (DMNV), which serves as the origin of the main efferent fibers that innervate the gastrointestinal tract [79]. 

The NTS acts as a relay station for the afferent information on its way to the central automatic network (CAN), which encompasses key areas such as the paraventricular nucleus of the hypothalamus, locus coeruleus, parabrachial nucleus, and the limbic system. The CAN predominantly regulates the hypothalamic–pituitary–adrenal axis (HPA), which is responsible for the stress response, as well as the autonomic nervous system (ANS) [79]. 

Indeed, the efferent fibers of the VN constitute about 20% of its total composition. As mentioned earlier, these fibers arise from the DMNV and play a pivotal role in regulating the tone of the smooth muscle in the gastrointestinal tract. These efferent fibers engage in two distinct pathways with contrasting effects on gut function. The excitatory pathway comprises cholinergic preganglionic efferent neurons located within the intermuscular plexus of the enteric nervous system (ENS). Ultimately, this pathway stimulates gut movement, promoting peristaltic and enhanced motility of the digestive tract.

Conversely, the inhibitory pathway involves cholinergic preganglionic neurons that emerge from the caudal DMNV. These preganglionic neurons synapse with non-cholinergic postganglionic neurons that release signaling molecules such as adenosine triphosphate (ATP), nitric oxide (NO), and vasoactive intestinal peptide (VIP). This pathway results in the relaxation of the smooth muscles and the dilatation of blood vessels, contributing to the inhibition of gut motility and a reduction in the tone of the gastrointestinal tract [83].

Vagal afferents are not in direct contact with gut microbiota or their intraluminal compounds under normal circumstances, unless there is a pathological alteration in the integrity of gut epithelium [79]. However, the VN can sense and relay signals from microbiota to the brain through mechanisms involving enteroendocrine cells (EECs) [84]. Firstly, when SCFAs bind to their respective receptors on EECs, these cells release various mediators: serotonin, GLP-1, PYY, and CCK. These mediators then act on receptors within VN, influencing various autonomic responses. These responses include modulation of gut motor and secretory functions, inflammatory response regulation, and mucosal defense mechanism enhancement [85]. Secondly, as previously mentioned, EECs can form direct synaptic connections with vagal afferent fibers, giving rise to specialized structures known as neuropods. These neuropods play a crucial role in mediating communication between the gut and the brain, contributing to the intricate brain–gut axis network [86]. Lastly, EECs are the first cells to encounter intraluminal components, allowing them to release various molecules such as hormones, neurotransmitters, and metabolites. The release of these molecules is guided by signals received from gut microbiota, further highlighting the dynamic interaction between EECs and the microbiota [84].

EECs are equipped with various types of receptors on their surface that allow them to interact with bacterial structural components or metabolites. For instance, Toll-like receptors (TLRs) are present for recognizing microbe-associated molecular patterns (MAMPS), including lipopolysaccharide (LPS), while G protein-coupled receptors (GPCRs) are responsible for binding SCFAs. These interactions play a role in stimulating vagal afferents and relaying signals to the brain [87,88]. Furthermore, the enteric neurons that constitute the enteric nervous system (ENS), often referred to as the “second brain” of the body, can sense both structural and chemical disturbances within the gastrointestinal tract. This sensory information can lead to the activation of the vagus nerve (VN) [84].

Additionally, when there is impairment in the integrity of the intestinal barrier, bacterial antigens from the gut lumen can trigger an immune response involving immune cells; these activated immune cells release pro-inflammatory molecules, such as TNF-α. Vagal afferents contain cytokine receptors for cytokines, allowing them to detect the inflammatory state of the gut. They then transmit signals to the CNS, initiating an anti-inflammatory response through a pathway known as the cholinergic anti-inflammatory pathway. This pathway helps to regulate and mitigate inflammation [89]. Indeed, apart from indirect stimulation of VN through the intricate interactions among EECs, ENS, and the gut immune system, it has been demonstrated that vagal afferents can also be directly activated by gut microbiota and their respective metabolites. Lal et al., have provided evidence that SCFAs, particularly butyrate, can directly stimulate vagal afferents in a manner independent of CCK [90].

In addition to their direct and indirect effects on the VN, SCFAs also regulate normal brain function and structure through various neuro-humoral pathways. The endothelial cells forming the BBB are equipped with monocarboxylate (MCTs) on their surfaces, allowing SCFAs to easily penetrate the BBB. Research indicates that both butyrate and propionate can enhance BBB integrity through distinct mechanisms [91].

These SCFAs not only enhance the expression of tight junction proteins, such as Occludin, claudin-5, and zona occludens-1, which are crucial for maintaining BBB integrity, but they also reduce the inflammatory response to microbial infections within the brain [92]. Furthermore, SCFAs stimulate the nuclear factor erythroid 2-like 2 (NFE2L2) pathway, an important component in protecting against inflammation-induced oxidative damage [93].

In addition to neurotransmitter modulation, SCFAs also regulate the level of neurotrophic factors essential for proper neuronal and neural network development in both the central and peripheral nervous systems. These factors, including nerve growth factor (NGF), glial cell line-derived neurotrophic factor (GDNF), and brain-derived neurotrophic factor (BDNF), contribute to functions such as memory formation and learning abilities [94,95,96].

A decrease in vagal tone attributed to dysautonomia has been observed in the context of dysbiosis associated with irritable bowel syndrome. The assessment of vagal tone can serve as an indicator of the status of the microbiota–gut–brain axis. Strategies aiming to enhance vagal tone through methods like vagus nerve stimulation and interventions to modulate the microbiota such as prebiotics, probiotics, dietary changes, and drugs targeting the cholinergic system could hold promise for restoring equilibrium within the microbiota–gut–brain axis [97]. 

SCFAs and long-chain fatty acids can stimulate vagal afferents in the intestines through distinct mechanisms. Specifically, butyrate exerts a direct effect on terminal afferent fibers. Since vagal nerve fibers do not extend into the intestinal lumen, the action of SCFAs occurs immediately after absorption through the epithelial mucosa, reaching the nerve endings situated in the lamina propria [90].

To elucidate the impact of acetate on the GBA, Peri et al., conducted an experimental study involving rats subjected to a high-fat diet. Their findings revealed that parasympathetic nervous system activation in these subjects resulted in heightened secretion of ghrelin and glucose-stimulated insulin release. This phenomenon can be characterized as a detrimental cycle that predisposes individuals to hyperphagia, perpetuating the loop of excessive food intake. Furthermore, the study identified additional consequences, including hypertriglyceridemia, increased accumulation of lipids in the liver and the skeletal muscles, and elevated insulin resistance within these regions. These observations underscore the notion that exposure to a hypercaloric diet with surplus food consumption fosters the development of obesity and its associated complications, potentially mediated by the actions of the brain–gut–microbiota axis [98].

Increases in appetite and growth hormone appear to be instigated by the influence of ghrelin on the central nervous system. Initially, it was hypothesized that ghrelin could traverse the BBB; however, subsequent investigations have shown that signals propagated via the afferent pathway of the vagus nerve might also serve as a route through which peripheral ghrelin communicates with the brain. Notably, interference with the afferent vagal pathway dampens both the ghrelin-induced feeding response and the release of growth hormone induced by ghrelin. Moreover, the modulation of the firing frequency of vagal afferent fibers after ghrelin administration underscores the direct correlation between ghrelin activity and the afferent vagal pathway [99].

The administration of acetate, propionate, and butyrate, along with the consumption of fermented carbohydrates, has been observed to curb food intake in the short term. While the exact mechanism underlying this phenomenon is not yet fully understood, SCFAs apparently achieve this by activating vagal afferents, transmitting information from the gastrointestinal tract to the brain. Moreover, the intraperitoneal injection of SCFAs leads to the suppression of food intake, with the potency of this effect varying based on the specific fatty acid injected, wherein butyrate demonstrates the most robust effect, followed by propionate and then acetate. This effect on food intake is dampened upon performing hepatic branch vagotomy, and it is entirely abolished when administering capsaicin, indicating that the vagus nerve serves as the mediator through which SCFAs influence food intake [100].

Chronic treatment with Lactobacillus strains has been shown to induce alterations in the expression of GABA receptors within specific brain regions, particularly cortical regions. Notably, these changes in receptor expression were accompanied by reductions in the hippocampus and amygdala when compared to control mice. Moreover, the observed behavioral and neurochemical effects were not found in mice that underwent vagotomy, highlighting the significance of the vagus nerve as a pivotal modulator of the communication between the gut and the brain. These findings suggest that certain microorganisms have the potential to offer assistance in addressing stress-related conditions such as depression and anxiety [101]. In instances of chronic colitis, which is linked to anxiety-like behavioral disturbances, these symptoms were not present in mice that had undergone vagotomy.

Furthermore, the administration of probiotics containing Bifidobacterium longum led to normalizing these behavioral patterns. Importantly, the anxiolytic effect was absent in mice experiencing anxiety disorders that had also undergone vagotomy. These observations underscore the vagus nerve’s role in mediating the gut microbiota’s effects on mood behavior [102].

An interesting observation has been made regarding BBB permeability in the context of germ-free mice. Germ-free mice exhibit heightened BBB permeability compared to mice harboring normal gut microbiota. This increased permeability in germ-free mice is associated with a reduction in the expression of tight junction proteins, including Occludin and Claudin 5, which are essential for maintaining the barrier function of the BBB. However, exposure of germ-free mice to normal microbiota has been found to mitigate BBB permeability, concurrently increasing the expression of these critical tight junction proteins. This suggests a potential role for gut microbiota in regulating BBB integrity [92].

Serotonin, derived from intestinal enterochromaffin cells, activates the 5-HT3 and 5-HT2 receptors on vagal afferents to mediate the stimulation of pancreatic secretion. Vagal electrical stimulation induces pancreatic secretion, and administration of 5-HT3 and 5-HT2 receptor antagonists did not affect the pancreatic response when the vagus nerve was electrically stimulated [103].

Butyrate has been demonstrated to benefit mice exposed to chronic stress. It effectively counteracts depressive behavior induced by prolonged stress exposure. Additionally, butyrate administration leads to an elevation in serotonin levels within the hippocampus and an increase in the expression of brain-derived neurotrophic factor (BDNF). Notably, it also appears to modulate the levels of Occludin, suggesting its potential to rectify any defects that may be present within the BBB [104].

Although the uptake of SCFAs by the brain is limited, their passage across the BBB allows them to influence neurotransmitter and neurotrophic factor levels [9,10]. SCFAs have been found to reduce the concentration of glutamine, glutamate, and GABA in the hypothalamus while increasing the level of anorexigenic neuropeptides [105]. Moreover, SCFAs play a role in modulating the synthesis of neurotransmitters like serotonin, dopamine, noradrenaline, and adrenaline by promoting the expression of enzymes such as tryptophan 5-hydroxylase 1 and tyrosine hydroxylase [106,107].

A decrease in hippocampal neurogenesis and memory retention was observed in adult mice treated with antibiotics. Reconstitution with normal intestinal flora did not reverse the deficits unless the mice received probiotics. Additionally, there was an increase in the number of Ly6Chi monocytes. Depleting these monocytes using antibodies decreased neurogenesis, while the transfer of monocytes restored neurogenesis after antibiotic treatment [108].

### 3.6. Short-Chain Fatty Acids’ Involvement in Amyotrophic Lateral Sclerosis Pathogenesis

An undeniable relationship between microbiota-induced disturbances in intestinal homeostasis and the neurological system has already been described in various neurodegenerative disorders, such as Alzheimer’s disease, Parkinson’s disease, and multiple sclerosis [109]. It is clear that microbiota metabolites, namely SCFAs, can modulate the gut–brain axis. However, the exact mechanisms underlying their importance in ALS pathogenesis and their therapeutic potential among ALS treatment options are still under investigation. 

ALS is a devastating disorder primarily characterized by neurodegeneration and muscular weakness. However, Rowin et al. [110] demonstrated that gastrointestinal symptoms, such as prolonged gastric emptying and constipation, might precede neurological symptoms. Studies conducted on both human subjects and SOD1G93A mouse models have shown that ALS is associated with gut dysbiosis, leading to decreased amounts of beneficial bacteria and a detrimental shift in the microbial profile. Consequently, there is an increase in intestinal inflammation and permeability [110,111,112]. For instance, Wu et al. [111] reported abnormally high intestinal permeability due to a considerable reduction in the expression of tight and adherens junction proteins (zonula occludens-1 and E-cadherin, respectively). This gut leakage leads to disturbances in microbiome homeostasis by decreasing the beneficial bacterial metabolites (e.g., SCFAs) while increasing the circulating toxic products (e.g., LPS), which eventually results in monocyte activation [113]. There is also an augmentation in systemic inflammation, as indicated by increased levels of intestinal and serum IL-17 [111]. Both the BBB and brain–spinal cord barrier (BSCB) exhibit abnormal permeability in ALS patients, allowing spinal cord and brain penetration with immune cells (e.g., macrophages, mast cells) and increased expression of COX2 [113,114].

Furthermore, they also observed an increased number of pathological Paneth cells, which are key components of the gut’s innate immune response responsible for releasing antimicrobial peptides (AMPs) and shaping the intestinal microbial profile through AMPs. In SOD1G93A mouse models, a significant reduction is seen in both AMP defensin 5 alpha and in butyrate-producing bacteria (e.g., *Butyrivibrio fibrisolvens*, *Escherichia coli*, *Firmicutes*), and this occurs before the onset of ALS [111].

Brenner and colleagues studied the fecal microbiome in individuals with ALS. Their findings indicated that distinctions between ALS patients and those in good health were evident solely in terms of the number of microbial species and the prevalence of uncharacterized *Ruminococcaceae*, with no significant change in the gut microbiota composition in ALS patients [115]. 

However, Di Gioia’s research revealed that ALS is linked to fluctuations in certain gut microbial elements when compared to control subjects, even in patients with minimal disability. This study has illustrated that the composition of the gut microbiota undergoes modifications throughout the progression of the disease, as evidenced by notable changes in specific microbial groups during the follow-up period. An imbalance has been observed between potentially beneficial microbial groups, like *Bacteroides*, and those with the potential for neurotoxic or pro-inflammatory effects, such as *Cyanobacteria* [116]. In a similar investigative vein, Zeng (2020) analyzed fecal community diversity and revealed an obvious alteration in the microbial composition of the gut microbiome in ALS patients. Specifically, at the phylum level, *Bacteroidetes* exhibited an increase, whereas *Firmicutes* (with protective effects) demonstrated a decrease when compared to healthy controls [117].

The C9orf72 gene mutation stands out as one of the prevalent genetic mutations linked to familial ALS. This mutation results in the production of abnormal RNA and protein aggregates within cells, which can contribute to the process of neurodegeneration [118,119]. A recent study by Burberry et al., has uncovered a potentially significant influence of the microbiome on the initiation and progression of symptoms in patients carrying C90RF72 mutations. This research supplies compelling evidence that the composition of the gut microbiome plays a crucial role in maintaining brain health, and can engage in unexpected interactions with established genetic risk factors associated with neurodegenerative disorders [120].

Zhang et al. [112] focused their research on 2% butyrate diet supplementation in G93A ALS mice, and have demonstrated a significant improvement in gut structure-enhanced intestinal barrier integrity and function, including a decreased percentage of abnormal Paneth cells, as well as reduced aggregation of G93A superoxide dismutase 1 mutated protein. A few years later, Zhang et al. [109] demonstrated that manipulating the microbiome through butyrate and antibiotic administration also improves muscle performance in murine ALS models, delays disease onset, and might even prolong patients’ lifespan. Ryu et al. [121] showed that phenylbutyrate activates transcriptional and post-translational molecular pathways that support motor neuron survival and alleviate ALS progression in G93A ALS mice.

According to various studies that have tested its efficiency, butyrate possibly possesses characteristics that can improve ALS prognoses. These include the inhibition of histone deacetylation, which prevents abnormal genes from being turned on; the reduction of neuroinflammation through the effects mentioned above of SCFAs on innate and adaptive immune response; and enhancement of energy metabolism by upregulating genes associated with mitochondrial metabolism [122]. Only two clinical trials have examined the efficiency of butyrate administration on ALS progression. The first trial, conducted by Cudkowicz et al. [123], involved 40 ALS patients treated with increasing doses of sodium phenylbutyrate (NaPB) for 20 weeks. This trial showed no clinical improvement compared to general findings on ALS development. The second trial was a randomized, double-blind, placebo-controlled phase 2 study that included 89 ALS patients treated with a combination of NaPB and tauroursodeoxycholic acid (TUDCA), known as AMX0035, along with 48 ALS patients receiving placebo over 24 weeks [122,124,125]. Disease evolution and survival were better in the first group; however, it is impossible to conclude which of the two components of the treatment contributed to these results [13,124,125]. A randomized, double-blind, placebo-controlled phase 3 trial to evaluate the safety and efficacy of AMX0035 for ALS treatment is currently ongoing [126].

Nonetheless, the appropriate dosage and formula of butyrate are yet to be decided. Cudkowicz et al. [123] concluded that 9 g per day is sufficient to modulate the histone deacetylation pathway, and further dose increases are pointless. The desired form in which to administer butyrate has to minimize possible adverse events (dizziness, diarrhea, nausea, dry mouth, peripheral edema, rash, fatigue, anxiety, abdominal pain) while maximizing patients’ tolerance of the drug [122]. Butyrate itself has a sour milk taste, is rapidly absorbed in the upper gastrointestinal tract, and poorly penetrates the BBB; therefore, it can be delivered via enemas, microencapsulated formulations, or cation-conjugated molecules (e.g., NaPB) [122,127,128].

The brain and spinal cord are highly insulin-sensitive organs, and glucose intolerance with insulin resistance has been linked to ALS [129]. The agonist for the GLP-1 receptor increases insulin levels, consequently lowering blood glucose levels. Insulin plays an important role in CNS-mediated regulation of glucose and energy homeostasis. Manzo et al. demonstrated in murine TDP-43 ALS models that upregulation of glycolysis by increasing motor neuron expression of phosphofructokinase and GLUT3 acts to exert neuroprotective effects [130].

Incretin hormone GLP-1 receptors (GLP-1R) are widely expressed on neurons and reactive glial cells in the spinal cord and the brain. Activation of GLP-1R exerts neurotrophic and neuroprotective effects in neurodegenerative murine models [131]. Li et al. demonstrated in a SOD1 G93A ALS murine model that treatment with exedin-4 (Ex-4), which is a long-acting agonist of GLP-1R, improved motor neuronal viability after hydrogen peroxide-induced oxidative stress in NSC19 neurons. This treatment also reduced caspase-3-positive neurons in the lumbar spine and improved motor clinical parameters [132].

In another study, Keerie et al. showed that using the GLP-1 receptor agonist Liraglutide in a SOD1 G93A ALS mouse model did not influence the astrocyte reactivity, as indicated by glial fibrillary acidic protein (GFAP) staining. This agonist did not alter the progression of ALS, but it might influence synaptic connectivity and plasticity, as demonstrated in other neurodegenerative models [133,134].

## 4. Conclusions

Growing evidence suggests that SCFAs, the gut microbiota, and environmental factors may collectively influence neurodegenerative processes through various interconnected pathways, including immune function, anti-inflammatory effects, gut barrier function, and energy metabolism.

It is important to emphasize that the interaction between environmental factors, gut microbiota, SCFAs, and neurodegeneration is complex, and our understanding of the pathways contributing to neurodegenerative diseases is still incomplete. External factors such as dietary nutrients and toxin or pollutant exposures can affect the composition of gut microbiota and the production of SCFA. Dysbiosis and decreased SCFA production can influence the neurodegenerative process through complex signaling pathways involving the gut–brain axis. While compelling data supports the implication of the gut–brain axis and the microbiota in neurodegenerative pathologies, establishing direct causative links will require further research.

Adopting a healthy lifestyle, including a balanced diet rich in fiber and plant-based foods, may promote a beneficial and diverse gut microbiome, potentially impacting the risk of neurodegenerative diseases. Further research is required to comprehensively comprehend how specific interventions targeting the gut microbiome and SCFA production might influence neurodegeneration, particularly in ALS patients. Additionally, exploring whether these interventions could be developed into therapeutic strategies is necessary.

## Figures and Tables

**Figure 1 ijms-24-15094-f001:**
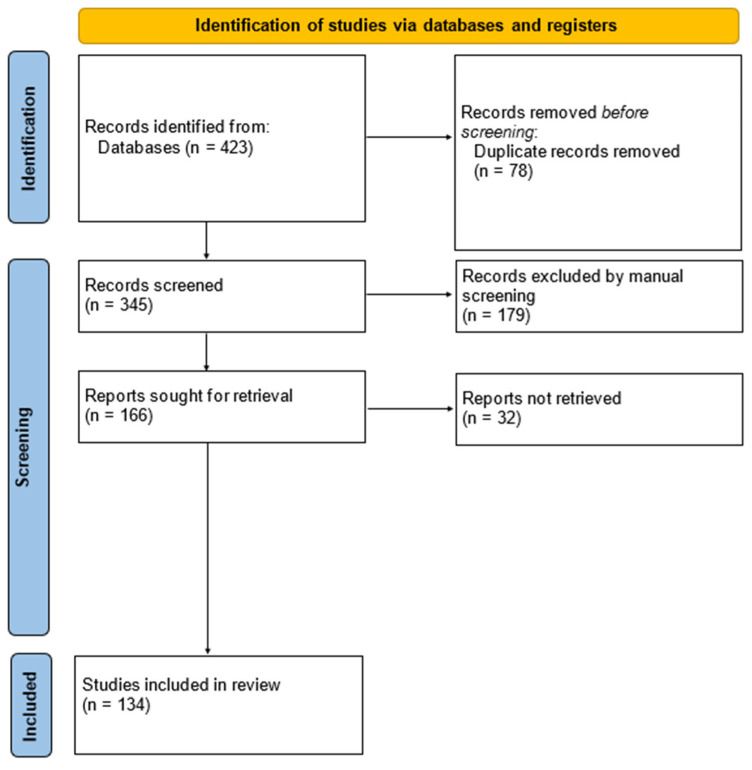
PRISMA flow diagram for the systematic review.

## Data Availability

Data is unavailable due to privacy.

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
