# Peer review of "The Role of Short-Chain Fatty Acids in Microbiota–Gut–Brain Cross-Talk with a Focus on Amyotrophic Lateral Sclerosis: A Systematic Review"

_ijms, 2023, doi:10.3390/ijms242015094_

Round 1
Reviewer 1 Report
The authors have undertaken an ambitious task of delving into the fascinating realm of the interplay between short-chain fatty acids, the gut microbiome, and neurodegenerative diseases, specifically ALS. They have adeptly summarized a complex body of literature spanning multiple disciplines. I appreciate the focus on potential implications for ALS, which serves to consolidate insights. However, I have a few specific comments:
1. The review primarily aims to explore the connections between short-chain fatty acids, the gut microbiome, and ALS. However, the authors dedicate an extensive portion of Section 3 to background information on short-chain fatty acids and the gut-brain signaling pathway, while providing comparatively less content related to ALS and the microbiome in Section 4. It is essential to rebalance the article by expanding the discussion on ALS and the microbiome. For instance, consider incorporating findings from patient studies exploring the relationship between microbiome composition and ALS, as well as examining the prevalence of gastrointestinal disorders in ALS patients.
2. The article focuses on ALS animal models, specifically SOD1-based murine models. To enhance comprehensiveness, the authors should also explore other ALS-related genes, such as C9orf72, FUS, and TDP43, which play crucial roles in the disease. Including information about these alternative models would provide a more comprehensive overview of the topic.
3. Lines 84-88, beginning with "3. Discussion...," appear to be erroneously included in the article and are unrelated to the central theme of this review.
Author Response
The authors have undertaken an ambitious task of delving into the fascinating realm of the interplay between short-chain fatty acids, the gut microbiome, and neurodegenerative diseases, specifically ALS. They have adeptly summarized a complex body of literature spanning multiple disciplines. I appreciate the focus on potential implications for ALS, which serves to consolidate insights. However, I have a few specific comments:
- The review primarily aims to explore the connections between short-chain fatty acids, the gut microbiome, and ALS. However, the authors dedicate an extensive portion of Section 3 to background information on short-chain fatty acids and the gut-brain signaling pathway, while providing comparatively less content related to ALS and the microbiome in Section 4. It is essential to rebalance the article by expanding the discussion on ALS and the microbiome. For instance, consider incorporating findings from patient studies exploring the relationship between microbiome composition and ALS, as well as examining the prevalence of gastrointestinal disorders in ALS patients.
The article's structure was modified, new data was added, and additional sections were created with the information being grouped by section and subsections.
- The article focuses on ALS animal models, specifically SOD1-based murine models. To enhance comprehensiveness, the authors should also explore other ALS-related genes, such as C9orf72, FUS, and TDP43, which play crucial roles in the disease. Including information about these alternative models would provide a more comprehensive overview of the topic.
Additional data was added regarding the genetic factor for ALS as suggested.
- Lines 84-88, beginning with "3. Discussion...," appear to be erroneously included in the article and are unrelated to the central theme of this review.
The typo was fixed and removed.
Reviewer 2 Report
The manuscript provides a systemic revision about the role of short-chain fatty acids in microbiota-gut-brain cross-talk, focusing also on the relationship between microbiota-induced disturbances in intestinal homeostasis and amyotrophic lateral sclerosis.
Overall, the manuscript is well written and reads well.
I have a major concern and some minor language points that the authors should address.
Major concern: lines from 84 to 88 contain a possible typo that must be removed
Minor language points:
1. Line 36: the expression “one of the most terrible neurodegenerative diseases” gives a subjective,(albeit realistic) interpretation of pathology severity and could be replaced with “a severe neurodegenerative disease”.
2. For a smoother reading, it would be advisable to remove some “the”, for example:
- line 52: “The gut microbiota”
- line 60: “The modulation”
- line 111: “the intestinal barrier”
- line 144: “between the microbiota and the immune system”
- line 152: “the cells' capacity”
- line 338: “the dietary fibers”
- line 513: “The increase in appetite and the secretion of growth hormone”, which could be substituted with “The increase in appetite and in growth hormone secretion”
- line 574: “the microbiota-induced disturbances”
- line 610: “the onset of the disease”, which could be substituted with “disease onset”
- lines 610-611: “the 610 lifespan of ALS patients”, which could be substituted with “ALS patients’ lifespan”
- lines 678-679: “the gut microbiota”
- line 682: “of the gut microbiota”
- line 685: “of the gut-brain axis and the microbiota”
3. Lines 471-472: the term “occluding” maybe refers to “occludin” and should be substituted.
4. The sentence “After screening, duplicate removals a number of 134 studies were identified and included into our review” is difficult to understand and should be revised.
5. Line 681: the term “exposure to toxins or pollutants” could be substituted with “toxin or pollutant exposures”.
See my comments above.
Author Response
We addressed all the issues that were pointed out, and further, we did a grammar check on the final version after the review changes.
Major concern: lines from 84 to 88 contain a possible typo that must be removed
We fixed and removed the typo.
Thank you for your suggestions!
Reviewer 3 Report
In this review, the authors discussed the Role Of Short-Chain Fatty Acids In Microbiota-Gut-Brain Cross-Talk with Focus On Amyotrophic Lateral Sclerosis. However, although the idea was good, the structure of the paper needs to be better. Namely, the authors should first decide what type of review they want to write (narrative review, scoping review, etc.) and in line with this write the new paper. In the method section, the authors presented a flowchart and showed that 134 papers were included in their review. Accordingly, they should have presented the results of these studies, further discussed them and, in the end, indicated future research directions in this field. Also, the authors should have highlighted a clear aim of this review.
Additionally, the authors have to cite all those authors whose results are mentioned in a review (some of them were missed in this review).
The authors could try to write some sentences more clearly.
Author Response
In this review, the authors discussed the Role Of Short-Chain Fatty Acids In Microbiota-Gut-Brain Cross-Talk with Focus On Amyotrophic Lateral Sclerosis. However, although the idea was good, the structure of the paper needs to be better. Namely, the authors should first decide what type of review they want to write (narrative review, scoping review, etc.) and in line with this write the new paper. In the method section, the authors presented a flowchart and showed that 134 papers were included in their review. Accordingly, they should have presented the results of these studies, further discussed them and, in the end, indicated future research directions in this field. Also, the authors should have highlighted a clear aim of this review.
As suggested, we tried to expand the results of some of the studies. We also changed the article's structure and followed a systematic review type. The aim of the study was also added to the manuscript to be more clear for the reader.
Additionally, the authors have to cite all those authors whose results are mentioned in a review (some of them were missed in this review).
We remade completely the references section, and missing citations were added.
Round 2
Reviewer 1 Report
The authors have addressed my questions. No more comments!
Reviewer 2 Report
None.
Reviewer 3 Report
Unfortunately, the authors did not answer my questions. This paper is not a systematic review, because the authors did not use guidelines addressing the introduction, methods, results, and discussion when preparing a systematic review. This paper does not have RESULTS and DISCUSSION and is not clear which papers the authors included in the text after the Introduction.
Minor editing of English language required